# Correlative Light and Electron Microscopy Using Frozen Section Obtained Using Cryo-Ultramicrotomy

**DOI:** 10.3390/ijms22084273

**Published:** 2021-04-20

**Authors:** Hong-Lim Kim, Tae-Ryong Riew, Jieun Park, Youngchun Lee, In-Beom Kim

**Affiliations:** 1Integrative Research Support Center, College of Medicine, The Catholic University of Korea, Seoul 06591, Korea; wgwkim@catholic.ac.kr (H.-L.K.); bloomwood@catholic.ac.kr (J.P.); leeycf@catholic.ac.kr (Y.L.); 2Department of Anatomy, College of Medicine, The Catholic University of Korea, Seoul 06591, Korea; trriew@gmail.com; 3Catholic Neuroscience Institute, College of Medicine, The Catholic University of Korea, Seoul 06591, Korea; 4Catholic Institute for Applied Anatomy, College of Medicine, The Catholic University of Korea, Seoul 06591, Korea

**Keywords:** correlative light and electron microscopy, immuno-electron microscopy, FluoroNanogold, cryo-ultramicrotomy

## Abstract

Immuno-electron microscopy (Immuno-EM) is a powerful tool for identifying molecular targets with ultrastructural details in biological specimens. However, technical barriers, such as the loss of ultrastructural integrity, the decrease in antigenicity, or artifacts in the handling process, hinder the widespread use of the technique by biomedical researchers. We developed a method to overcome such challenges by combining light and electron microscopy with immunolabeling based on Tokuyasu’s method. Using cryo-sectioned biological specimens, target proteins with excellent antigenicity were first immunolabeled for confocal analysis, and then the same tissue sections were further processed for electron microscopy, which provided a well-preserved ultrastructure comparable to that obtained using conventional electron microscopy. Moreover, this method does not require specifically designed correlative light and electron microscopy (CLEM) devices but rather employs conventional confocal and electron microscopes; therefore, it can be easily applied in many biomedical studies.

## 1. Introduction

Immuno-electron microscopy (Immuno-EM) is an experimental technique that is widely used in the biomedical research field, allowing the investigation of the ultrastructural distribution of specific target molecules. It can be performed in two ways depending on the embedding process: pre- and post-embedding [1,2,3,4,5,6,7]. The morphology and ultrastructure are well retained in the pre-embedding method, but deep penetration of the antibody is limited. The post-embedding method resolves this issue as the antigen-antibody reaction occurs on the surface of the ultrathin sectioned tissue [1,8,9]. Immunostaining of thawed ultrathin frozen sections using a high concentration of sucrose as a cryoprotectant, and low-temperature polymerization using hydrophilic resin are the key processes in post-embedding Immuno-EM [10,11,12]. These processes preserve the ultrastructure and antigenicity, but result in a lower contrast compared to the conventional EM image, and the laborious character of ultrathin sectioning constitute the disadvantages of the post-embedding method. Though conventional histo-cryostat sections have been used to overcome such problems, cryodamage hinders the observation of fine subcellular structures [13,14]. Because either method has clear advantages regarding the ultrastructural integrity and antigenicity, we have developed the method using cryo-ultramicrotomic semithin sectioning, which combined cryo-thick sectioning (0.5–2 μm) of vitrified samples and immunolabeling, as employed in post-embedding, with polymeric resin-embedding, as in the pre-embedding method. Using this technique, we were able to immunolabel two or more antibodies and to observe them using a light microscope and an electron microscope, with fine preservation of the ultrastructure, thus making the correlation between the two different modalities possible.

## 2. Results

### 2.1. Correlative Light and Electron Microscopy (CLEM) Procedures

Tissue or cells were fixed, and then moved to 30% sucrose and 2.3 M sucrose sequentially. Sucrose-infiltrated samples were mounted in 2.3 M sucrose on an aluminum sample holder (Leica, Wetzlar, Germany) and vitrified by immersion in liquid nitrogen. In order to reduce the folding and creasing, semithin cryosections (500 nm–2 µm-thick) instead of ultrathin cryosections of the vitrified samples were obtained at −90 °C using a glass knife or a diamond knife in a cryo-ultramicrotome (Leica EM UC7 ultramicrotome and FC7 cryochamber, Leica) (Figure 1A). Glass knives were replaced frequently to reduce cutting artifacts. Cryosections were picked up with a drop of 2.3 M sucrose in the loop and transferred onto a coated slide glass, which was then marked on the opposite side (Figure 1B). After immunostaining and confocal imaging (Figure 1C), the attached sections were post-fixed with 2.5% glutaraldehyde for 20 min and 1% osmium tetroxide in 0.1 M PB for 10 min. The sections were dehydrated using a graded ethanol series and then infiltrated with Epon 812 resin. The sections were embedded in resin-filled microcentrifuge tubes (0.5 mL) sealed with silicon rings (Figure 1D). After polymerization, the silicon rings were removed, and the slides were immersed in liquid nitrogen to remove the tubes. The region of interest was trimmed using a razor blade, and ultrathin serial sections of a thickness of 70–90 nm were obtained from the block face (Figure 1F,f,G,g1). Characteristic morphological features such as vasculatures, artifacts, and contours of the tissue sections were compared with confocal images to identify the region of interest (ROI) under transmission electron microscopy (Figure 1E,F). Sections were mounted on 100-mesh or formvar-coated one-hole-mesh copper grids (Figure 1G,g2), stained with 1% uranyl acetate and 0.02% lead citrate, and then observed using a transmission electron microscope (JEM 1010, JEOL, Japan) (Figure 1H).

### 2.2. Comparison between the Cryo-Ultramicrotomy Semithin Section and the Histo-Cryostat Section in Electron Microscopy

To examine whether cryodamage was induced in cryo-ultramicrotomy of rat brain tissues embedded in 2.3 M sucrose, we compared the electron microscopic image of the cryo-ultramicrotomy semithin sections with that of the conventional optimal cutting temperature (OCT) compound-embedded histo-cryostat sections (Figure 2). Conventional transmission electron microscopy (TEM) images at a lower magnification, taken from the hippocampus, evidenced ice crystal-induced cryodamage, such as empty spaces and vacuoles, which were frequently observed in the conventional cryostat sections (Figure 2A). At a higher magnification, the disruption of the intracellular contents and membranous organelles was observed. Although some postsynaptic density materials were detected, many neurites and synaptic structures were indistinguishable (Figure 2C). In contrast, in the semithin ultracryotomy sections embedded in 2.3 M sucrose, the tissue structures were clearly preserved without empty spaces induced by cryodamage, at a lower magnification (Figure 2B). Neurites and synaptic structures, such as the post-synaptic density and presynaptic vesicles, were well maintained at a higher magnification (Figure 2D). Thus, these results imply that cryo-ultramicrotomy semithin sections are suitable for electron microscopy, and are summarized in Table 1, showing differences in ultrastructure between conventional histo-cryostat sections and cryo-ultramicrotomic semithin sections.

### 2.3. CLEM of Cultured Cells and Brain Tissues

First, CLEM was used to test the cultured cell lines. Vitrified HEK293T cells transfected with GFP were semithin-sectioned and imaged using confocal microscopy (Figure 3A,B). The specific localization of GFP-expressing cells identified in the LM image correlated with the EM image, providing the ultrastructural details, including the nuclei, cell membranes, and intracellular organelles (Figure 3C,D).

To test the tissue samples, normal rat brains were double-labeled with NeuN, a neuronal nuclei protein that marks the nuclei and soma of neurons, and a 78 kDa glucose-regulated immunoglobulin heavy-chain binding protein (GRP78-Bip), a marker of the rough endoplasmic reticulum (rER). Light microscopic imaging showed GRP78-Bip expression in the cytoplasm around the NeuN (+) neuronal nuclei (Figure 4A,B). EM images were correlated with the ROI in the LM images, and the overlay of LM and EM images clearly showed the NeuN stained area in the nuclei of neurons (Figure 4C,D), and the GRP78-Bip stained area in the cytoplasm of neurons, where gold particles in the ER membrane confirmed the localization of GRP78-Bip in the ER of neurons (Figure 4E,F).

## 3. Discussion

Cryo-techniques are essential for maintaining both cell structure and antigenicity in Immuno-EM. However, technical difficulties related with the use of cryo-ultrathin sections (70–90 nm) and a limited viewing area hinder the general usage of such techniques [10]. In this study, we have modified the technique by using vitrified biological samples with semithin sections (0.52 µm) using a cryo-ultramicrotome. Vitrified samples embedded in a high-concentration sucrose solution retained tissue integrity and the cellular ultrastructure, while specific immunolabeling enabled us to correlate the LM images with the EM images.

Pre-embedding immunogold labeling using 10-μm-thick frozen sections, which are obtained using a histo-cryostat, has been reported previously [13,14]. Additionally, Kusumi et al. introduced CLEM using semithin (1 µm) cryosections at −65 to −75 °C in the backscattered electron (BSE)-mode scanning electron microscope [15]. We compared the OCT-embedded histo-cryostat samples cut at −20 °C with the 2.3 M sucrose-embedded samples cut at −100 °C, and we found that the embedding media and cutting temperature are critical factors in reducing the ice crystal-induced cryo-damage. In addition, sectioning at the nanometer scale using a diamond or glass knife reduced the distortion of the tissue sections compared to histo-cryostat tissue sections. However, high-pressure freezing (HPF) could be an alternative to achieve perfect vitrification, as HPF provides quick cooling, which is possible at a high pressure (200 MPa) at a deep depth (200 µm) [16].

Compared to pre-embedding preparation, no loss of fluorescent signal was observed because LM imaging was performed before the resin embedding step. Both GFP and immunofluorescence signals were evident enough to correlate with the electron microscopic findings. Furthermore, weak chemical fixation using paraformaldehyde, quick freezing in liquid nitrogen, and relatively thin (1–2 μm) sections made clear immunostaining possible. However, due to the pixel resolution limitations, the blurring of the fluorescent signals was a major difficulty in the correlation between LM imaging and EM imaging. FluoroNanogold (FNG) provides a precise correlation of the subcellular target proteins; for example, GRP-78-Bip is localized in the ER membranes in the EM images [17,18,19,20,21]. In addition, CLEM has been performed using consecutive sections of semithin and ultrathin sections, on which immunolabeling and LM observations were performed on the semithin sections and correlative electron microscopy was carried out on the consecutive ultrathin sections, which inevitably made the correlation of both images inaccurate [21]. In our protocol, however, we obtained every ultrathin section from the semi-thin-immunolabeled sample; thus, a more precise localization of the target proteins was possible [17,18,21,23]. Moreover, conventional resin embedding and positive staining using heavy metals after light microscopy provided better image contrast and resolution than when using Tokuyasu’s technique, in which the negative staining obscures the image contrast and the absence of the resin embedding process limits the spatial resolution [11].

In summary, we developed a novel CLEM by combining conventional TEM and confocal microscopy. Vitrification and semithin cryo-sectioning at −80–−100 °C provided a good retention of the ultrastructure without hindering antibody penetration. We expect this method to be widely used for localizing multiple target molecules at the ultrastructural level owing to its simplicity and efficiency.

## 4. Materials and Methods

### 4.1. Specimen Preparation

All procedures and provisions for animal care were in accordance with the Laboratory Animals Welfare Act, the Guide for the Care and Use of Laboratory Animals, and the Guidelines and Policies for Rodent Survival Surgery provided by the Institutional Animal Care and Use Committee (IACUC) at the College of Medicine, The Catholic University of Korea (Approval number: CUMS-2017-0321-05). IACUC and the Department of Laboratory Animals (DOLA) at the Catholic University of Korea, Songeui Campus, accredited the Korea Excellence Animal Laboratory Facility of the Korea Food and Drug Administration in 2017 and acquired the full Association for Assessment and Accreditation of Laboratory Animal Care (AAALAC) International accreditation in 2018. All efforts were made to minimize animal suffering and reduce the number of animals used. Adult male Sprague-Dawley rats (250–300 g, OrientBio, Seongnam, Korea) were divided into groups of three and each group was housed in a cage in a controlled environment at a constant temperature (22 ± 5 °C) and relative humidity (50 ± 10%) with food (gamma ray-sterilized diet) and water (autoclaved tap water) available ad libitum. They were maintained on a 12-h light/dark cycle.

Animals were deeply anesthetized with 10% zolazepam (20 mg/kg i.p.) and xylazine (7.5 mg/kg i.p.) and sacrificed via transcardial perfusion with 4% paraformaldehyde (PFA) in 0.1 M phosphate buffer (PB, pH 7.4). The brains were promptly removed and sliced to slides with a thickness of 1 mm, and the region of interest (ROI) was trimmed to 2 mm × 2 mm. After fixation in PFA for 3 h, the brains were moved to a 30% sucrose solution in PB for 3 h, and then moved to a 2.3 M sucrose solution in 0.1 M PB overnight at 4 °C for cryoprotection.

HEK293T cells were transfected with pEGFP-hXBP1 fusion protein-expressing plasmid, fixed in 4% PFA for 3 h, and then processed further in 30% sucrose and a 2.3 M sucrose solution in 1.5 mL conical tubes for cryoprotection.

### 4.2. Cryostat Sections for Electron Microscopy

Rat brains were fixed with 4% PFA and were infiltrated with 30% sucrose in 0.1 M PB overnight at 4 °C for cryoprotection. Tissues were mounted with optimal cutting temperature (OCT) compound and frozen by putting them onto steel plates which were kept in the liquid nitrogen for cooling. Coronal cryostat sections (10-µm-thick) were cut at −27 °C in a cryostat and placed on a coated slide and further processed for post-fixation and Epon polymerization, as described above.

### 4.3. Immunochemistry and Light Microscopy

Semithin cryosections were blocked with 10% normal goat serum, and then double-labeled using a mixture of rabbit polyclonal antibody to GRP78-Bip (1:2000; Abcam, Cambridge, UK) and mouse monoclonal antibody to NeuN (1:500; Millipore, Darmstadt, Germany) in 0.01 M PBS at 4 °C overnight. After washing with 0.01 M PBS, the sections were incubated in a mixture of Alexa Fluor 488-FNG goat anti-rabbit (1:100; Nanoprobes; Yaphank, NY, USA) and Cy3-conjugated goat anti-mouse antibody (1:2000, Jackson ImmunoResearch, West Grove, PA, USA). The sections were counterstained with DAPI for 10 min. PBS buffer was used for the mounting medium to remove the coverslips easily after imaging using a confocal microscope (LSM 800; Carl Zeiss Co. Ltd., Oberkochen, Germany). Light microscopic imaging was performed at various magnifications using differential interference contrast (DIC) settings (Figure 3A). High magnification (800× to 1260×) and high resolution (at least 1024 × 1024 pixels) are required, as the electron microscopy covers only a small portion of the light microscopic images. After fixation with glutaraldehyde and osmium tetroxide, silver enhancement was performed using an HQ silver enhancement kit (Nanoprobes) for 3 min.

### 4.4. CLEM and Image Processing

The specific ROIs in electron microscopy were localized and matched with the confocal images using DIC and DAPI channel images. Distinct morphological features, such as the contour of the tissue, vascular lumens, folded areas, and/or the distribution pattern of the cell nuclei, were used (Figure 1E,F). Using Photoshop (Adobe, San Jose, CA, USA), LM and EM images were overlapped in different layers, and the size and orientation of the images were adjusted to fit each other. The opacity of the overlapping layer was adjusted to show the correlation between the LM and EM images.

## Figures and Tables

**Figure 1 ijms-22-04273-f001:**
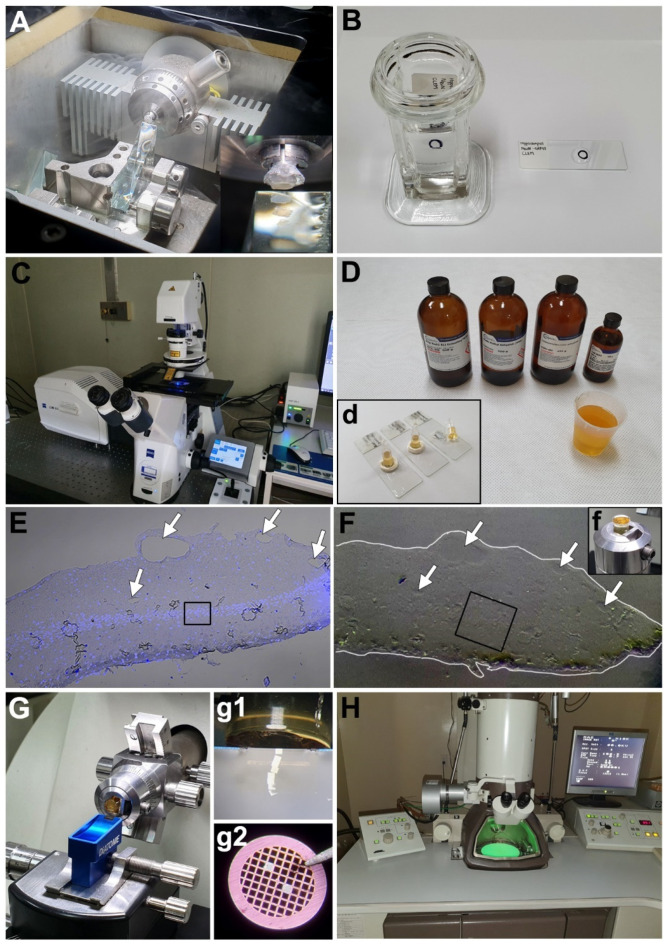
Correlative light and electron microscopy (CLEM) procedures on thick cryo-sections from the hippocampus using FluoroNanogold. (**A**) Cryoprotected sample with 2.3 M sucrose is cut on a cryo-ultramicrotome with a glass knife at −90 °C (1-μm-thick sections). (**B**) The sections are mounted on a glass slide with the loop containing 2.3 M sucrose and incubated with the antibodies for immunocytochemistry. (**C**) Light-microscopic imaging is performed using a confocal microscope. (**D**,**d**) After fixation and silver enhancement, the sections are processed for routine transmission electron microscopy (TEM) preparation using a Poly/Bed 812 embedding media using microcentrifuge tubes. (**E**) Confocal image (**E**) with DAPI staining and differential interference contrast (DIC) settings shows clear contour of the tissue and vascular lumens (arrows). The boxed area in E indicates the region of interest. (**F**) Embedded block (**f**) was observed under a stereoscope for identification of morphological cues for alignment. The contour of the tissue block face is delineated in white line, and identical vascular structures seen in E are marked with arrows. The boxed area in F shows a fine trimming area. (**G**,**g1**) Polymerized block is trimmed and cut on an ultramicrotome with a diamond knife (80-nm-thick slices) without a semithin section. (**g2**) The section is mounted on a grid (100-mesh) and stained with uranyl acetate. (**H**) Region of interest imaged using light microscopy is observed using TEM.

**Figure 2 ijms-22-04273-f002:**
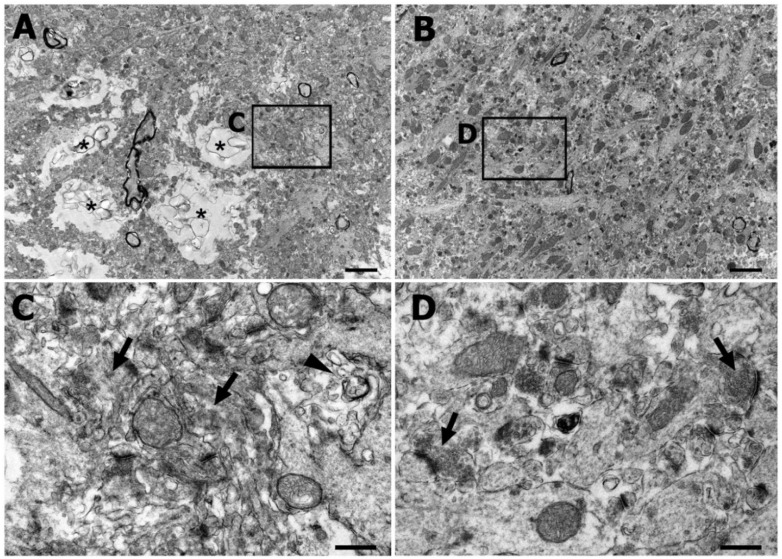
Comparison between the conventional transmission electron microscopy images of histo-cryostat sections and semithin cryo-sections, using a cryo-ultramicrotome, of rat hippocampus. (**A**) Lower magnification view of the histo-cryostat section showing empty spaces presumably induced by cryodamage (asterisks). (**B**) Lower magnification view of the semithin cryo-section showing a well-preserved tissue structure. (**C**) Higher magnification view of the boxed area in A showing damaged intracellular organelles (arrowheads) and obscure neurites (arrows). (**D**) Higher magnification view of the boxed area in B showing better outlined neuronal processes and synaptic vesicles (arrows). Scale bar = 2 μm for (**A**,**B**); Scale bar = 0.5 μm for (**C**,**D**).

**Figure 3 ijms-22-04273-f003:**
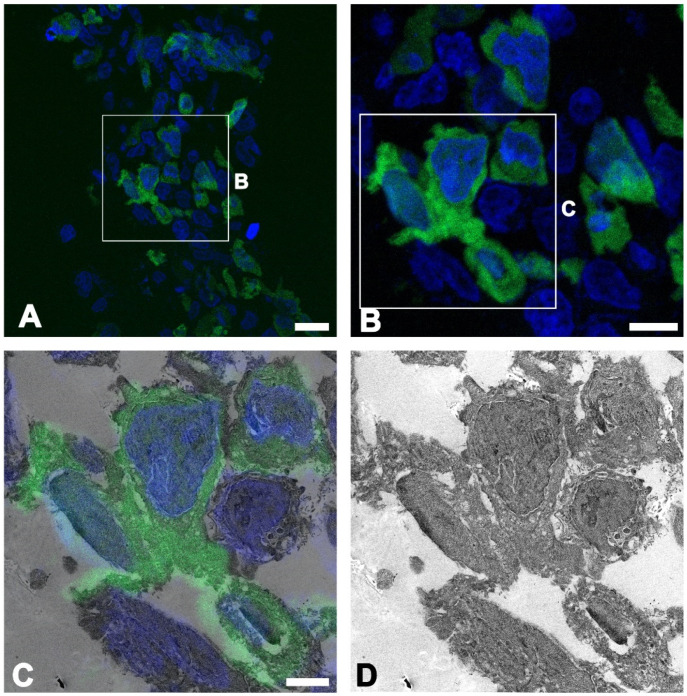
Correlated fluorescent signal and transmission electron microscopy image of the pEGFP-expressing HEK293T cells. (**A**,**B**) Confocal microscopic image showing the fluorescent signal in the cytoplasm of the HEK293T cells. (**C**,**D**) Confocal microscopic image overlaid onto the corresponding electron microscopic image, which can distinguish GFP-expressing cells from non-GFP-expressing cells with a detailed ultrastructure. Scale bar = 20 µm for (**A**); Scale bar = 10 µm for (**B**); Scale bar = 5 µm for (**C**,**D**).

**Figure 4 ijms-22-04273-f004:**
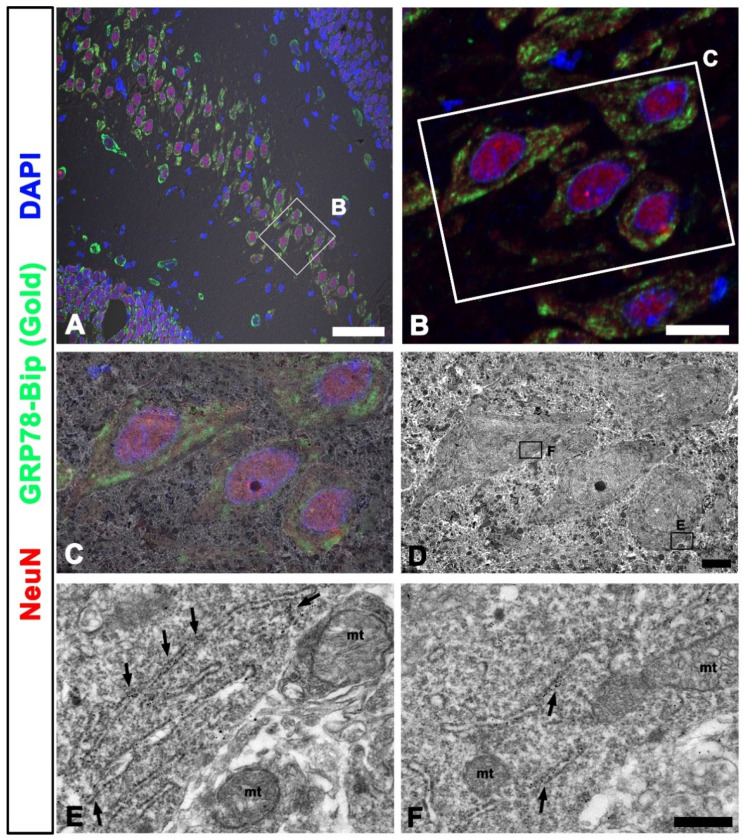
Correlated fluorescent signal and TEM image in the rat hippocampus. (**A**,**B**) Double-labeling of GRP78 and NeuN on confocal microscope images showing prominent GRP78-Bip signals in the vicinity of NeuN (+) neuronal nuclei. (**C**) Confocal microscopic image overlaid onto the corresponding electron microscopic image. (**D**) The corresponding transmission electron microscopic images obtained from the same field. (**E**,**F**) Higher magnification of the boxed area showing the gold particles in the rough endoplasmic reticulum (rER) membranes (arrows) coinciding with the green signals of GRP78-Bip in (**A**,**C**). mt, mitochondria. Scale bar = 50 µm for (**A**); Scale bar = 10 µm for (**B**); Scale bar = 4 µm for (**C**,**D**); Scale bar = 0.4 µm for (**E**,**F**).

**Table 1 ijms-22-04273-t001:** Comparison between ultrastructure of histo-cryostat section and cryo-ultramicrotomic semithin section.

Cellular Components	Histo-Cryostat Section	Cryo-Ultramicrotomic Semithin Section
Cell membrane	Frequently disrupted (empty spaces and vacuoles)	Well preserved
Subcellular organelles	Frequently disrupted	Well preserved
Synaptic structures (presynaptic vesicles and postsynaptic density)	Indistinguishable, frequently disrupted	Well preserved

## Data Availability

All relevant data are included within the manuscript.

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
