# Peer review of "Correlative Light and Electron Microscopy Using Frozen Section Obtained Using Cryo-Ultramicrotomy"

_ijms, 2021, doi:10.3390/ijms22084273_

Round 1
Reviewer 1 Report
In this study, Kim et al developed a method by combining cryo-thick sectioning and immunolabeling to enable CLEM imaging. The technique presented here has important applications in biomaterials imaging. The methods and results are convincing and clearly presented. Some minor revisions are recommended.
1. This is a method paper, so the method introduction should be a major part of the text. I suggest moving Fig.4 to Fig. 1, and highlight the methods and materials/instruments in the Results part of the text.
2. Cryo-ultramicrotomy are known to cause cutting artifacts to the specimens. How did the authors overcome this problem?
3. Can the authors comment on the staining efficiency in their semithin cryosection staining?
Author Response
We are grateful to the editor and reviewers for giving their valuable time to evaluate our manuscript and providing helpful comments. Their feedback, which we found to be very reasonable, has helped us to improve our manuscript significantly. Please find our point-by-point responses (in regular font) to the reviewers’ comments (in italics) below. The revised sections of the manuscript are indicated in red font.
Reviewer 1
In this study, Kim et al developed a method by combining cryo-thick sectioning and immunolabeling to enable CLEM imaging. The technique presented here has important applications in biomaterials imaging. The methods and results are convincing and clearly presented. Some minor revisions are recommended.
- This is a method paper, so the method introduction should be a major part of the text. I suggest moving Fig.4 to Fig. 1, and highlight the methods and materials/instruments in the Results part of the text.
Response: We appreciate and agree with this suggestion. As per the reviewer’s suggestion, we have added the following text to the Results section, and revised the materials/methods part of the text. In addition, we have moved the Figure 4 to Figure 1, and added new figures to Figure 1 and the corresponding figure legends.
In results: “2.1 Correlative light and electron microscopy (CLEM) procedures
“Tissue or cells were fixed, and then moved to 30% sucrose and 2.3 M sucrose sequentially.” (page 2, lines 52-54)
"Characteristic morphological features such as vasculatures, artifacts, and contours of the tissue sections were compared with confocal images to identify the region of interest (ROI) under transmission electron microscopy (Fig 1E, F).” (page 2, lines 69-72)
In figure legends: “(E) Confocal image (E) with DAPI staining and DIC setting shows clear contour of the tissue and vascular lumens (arrows). The boxed area in E indicates the region of interest. (F) Embedded block (f) was observed under a stereoscope for identification of morphological cues for alignment. The contour of the tissue block face is delineated in the white line, and identical vascular structures seen in E are marked with arrows. The boxed area in F shows a fine trimming area.” (page 3, lines 83-87)
- Cryo-ultramicrotomy are known to cause cutting artifacts to the specimens. How did the authors overcome this problem?
Response: We are thankful for the point raised by the reviewer. Because cryo-ultramicrotomy usually gives rise to cutting artifacts such as folding and crease, due to the very thin (< 200 nm) nature of the tissue sections, we made the tissue sections relatively thicker (between 1 and ~2μm) and trimmed this sample later on after resin embedding. In addition, the glass knife was replaced frequently. Therefore, we have added the following text regarding these issues in the revised manuscript.
“In order to reduce the folding and crease, semithin cryosections (500 nm–2 µm-thick) instead of ultrathin cryosections, of the vitrified samples were obtained at −90°C using a glass knife or a diamond knife in a cryo-ultramicrotome (Leica EM UC7 ultramicrotome & FC7 cryochamber, Leica) (Fig. 1A). Glass knives were frequently replaced to reduce cutting artifacts.” (page 2, lines 55-60)
- Can the authors comment on the staining efficiency in their semithin cryosection staining?
Response: We are grateful for this suggestion by the reviewer. Clear immunostaining was possible as our method employs the weak chemical fixation, quick freezing, and thin tissue sections. Therefore, we have added the following text to the Discussion section.
“Furthermore, weak chemical fixation using paraformaldehyde, quick freezing in liquid nitrogen, and relatively thin (1–2 μm) sections provided clear immunostaining possible.” (page 7, lines 176-178)
Reviewer 2 Report
Immuno-electron microscopy is a useful tool in cell biology. The authors combined confocal fluorescent microscopy with transmission electron microscopy and apply these approaches to observe both cell and tissue. They demonstrate that the combined approach is superior to conventional immuno-electron microscopy. I have the following comments.
In the introduction, the authors would have introduced both the cryo-ultramicrotomy semithin section and the histo-cryostat section in electron microscopy,
Line 40, on the contrary, the negative staining enhances the contrast of an image.
Line 50, It is obvious that the cryo-ultramicrotomy semithin sections have fewer artifacts if compared with the histo-cryostat section. Because the one was performed at -90 °C and the histo-cryostat section was performed at -27 °C.
Line 58, How frequently the authors observe the feature of cryo-damage? For the case of the semithin section, how often do you observe the cryo-damage? Concerning the cryo-damage, making a comparison in a table would be helpful for the reader to follow.
Line 98, Figure 3E and 3F, Arrows stands for?
Line 128, results in
Line 192, When freezing the tissue in liquid nitrogen within -196 °C, then transferring it to -27 °C, a warming-up causes ice-crystal.
Line 222, the author would describe in detail how to make alignment between confocal images and TEM images since both images have been taken in different magnifications, resulting in different resolutions.
Author Response
We are grateful to the editor and reviewers for giving their valuable time to evaluate our manuscript and providing helpful comments. Their feedback, which we found to be very reasonable, has helped us to improve our manuscript significantly. Please find our point-by-point responses (in regular font) to the reviewers’ comments (in italics) below. The revised sections of the manuscript are indicated in red font.
Immuno-electron microscopy is a useful tool in cell biology. The authors combined confocal fluorescent microscopy with transmission electron microscopy and apply these approaches to observe both cell and tissue. They demonstrate that the combined approach is superior to conventional immuno-electron microscopy. I have the following comments.
- In the introduction, the authors would have introduced both the cryo-ultramicrotomy semithin section and the histo-cryostat section in electron microscopy,
Response: We appreciate the reviewer’s suggestion. We have added the following text and revised the following sentence.
In introduction: “Though conventional histo-cryostat sections have been used to overcome such problems, cryodamage hinders the observation of fine subcellular structures [13,14]” (page 1, lines 41-43).
“~, we have developed the method using cryo-ultramicrotomic semithin section, which combined cryo-thick sectioning (0.5 μm–2 μm) of vitrified samples and immunolabeling, as employed in post-embedding, with polymeric resin-embedding, as in the pre-embedding method.” (pages 1-2, lines 44-47).
- Line 40, on the contrary, the negative staining enhances the contrast of an image.
Response: We thank the reviewer for this comment. Immunostaining of thawed ultrathin frozen sections results in negative image that enhances the contrast of an image, but its contrast is lower than the usual positive staining images as shown in the results. In addition, the low-temperature polymerization method provides a positive image, but its contrast is lower than that of the conventional EM. Therefore, we have added and revised the following text to the revised manuscript.
“These processes preserve the ultrastructure and antigenicity, but result in a lower contrast compared to the conventional EM image, and the laborious character of ultrathin sectioning constitute the disadvantages of the post-embedding method.” (page 1, lines 38-41)
- Line 50, It is obvious that the cryo-ultramicrotomy semithin sections have fewer artifacts if compared with the histo-cryostat section. Because the one was performed at -90 °C and the histo-cryostat section was performed at -27 °C.
Response: Yes! In this study, we introduced a method using the cryo-ultramicrotomy semithin sections.
- Line 58, How frequently the authors observe the feature of cryo-damage? For the case of the semithin section, how often do you observe the cryo-damage? Concerning the cryo-damage, making a comparison in a table would be helpful for the reader to follow.
Response: We thank the reviewer for this valuable suggestion. In the semithin cryo-ultramicrotomic sections, we rarely observed any artifacts due to cryodamage, as shown in figure 2. In order to improve the readability of such findings, we have added the following text and Table 1 in the revised manuscript.
“Thus, these results imply that cryo-ultramicrotomy semithin sections are suitable for electron microscopy, and are summarized in Table 1 showing differences in ultrastructure between conventional histo-cryostat sections and cryo-ultramicrotomic semithin sections.” (page 4, lines 107-110)
Table 1. Comparison between ultrastructure of histo-cryostat section and cryo-ultramicrotomic semithin section.
|
|
Histo-cryostat section |
Cryo-ultramicrotomic semithin section |
|
Cell membrane |
Frequently disrupted (empty spaces and vacuoles) |
Well preserved |
|
Subcellular organelles |
Frequently disrupted |
Well preserved |
|
Synaptic structures (presynaptic vesicles and postsynaptic density) |
Indistinguishable, frequently disrupted |
Well preserved |
- Line 98, Figure 3E and 3F, Arrows stands for?
Response: We thank the reviewer for this comment. We have added the following in the figure legend:
In figure legend: “(E, F) Higher magnification of the boxed area showing the gold particles in the rER membranes (arrows) coinciding with the green signals of GRP78-Bip in A and C.” (page 6, line 151)
- Line 128, results in
Response: Unfortunately, we cannot understand what this comment means. Therefore, we apologize for not answering this question.
- Line 192, When freezing the tissue in liquid nitrogen within -196 °C, then transferring it to -27 °C, a warming-up causes ice-crystal.
Response: We agree with the reviewer’s concern. Some of the procedures were omitted in the description; therefore, we have changed the text in the revised manuscript as follows:
“Tissues were mounted with OCT compound and frozen by putting them onto steel plate which were kept in the liquid nitrogen for cooling.” (page 8, lines 225-226)
- Line 222, the author would describe in detail how to make alignment between confocal images and TEM images since both images have been taken in different magnifications, resulting in different resolutions.
Response: We appreciate the reviewer’s suggestion. For a detailed description of the alignment process, we have added Figure 1G and 1H in figure 1 and the corresponding figure legends. In addition, we have revised the text in the revised manuscript as follows:
In results: “Characteristic morphological features such as vasculatures, artifacts, and contours of the tissue sections were compared with confocal images to identify the region of interest (ROI) under transmission electron microscopy (Fig 1E, F).” (page 2, lines 69-72)
In figure legends: “(E) Confocal image (E) with DAPI staining and DIC setting shows clear contour of the tissue and vascular lumens (arrows). The boxed area in E indicates the region of interest. (F) Embedded block (f) was observed under a stereoscope for identification of morphological cues for alignment. The contour of the tissue block face is delineated in the white line, and identical vascular structures seen in E are marked with arrows. The boxed area in F shows a fine trimming area.” (page 3, lines 83-87)
In materials/methods: “High magnification (800x to 1260x) and high resolution (at least 1024 x 1024 pixels) are required, as the electron microscopy covers only small portion of the light microscopic images.” (page 8, lines 241-243)
“Distinct morphological features, such as the contour of the tissue, vascular lumens, folded areas, and/or the distribution pattern of the cell nuclei, were used (Fig. 1E-F).” (page 8, line 250)
Additionally, we found an incorrect description in Introduction section; thus, we fixed it, as follows:
“The morphology and ultrastructure are well retained in the pre-embedding method, but the penetration of the antibody is limited only to the surface of the sample section. Thus, the post-embedding method is more widely used for immunolabeling, because antigen-antibody reactions occur before embedding, enabling a deeper penetration of the antibody into the sample [1,8,9].” in introduction was changed into “The morphology and ultrastructure are well retained in the pre-embedding method, but deep penetration of the antibody is limited. The post-embedding method resolves this issue as the antigen-antibody reaction occurs on the surface of the ultrathin sectioned tissue [1,8,9].” (Page 1, lines 32-35)